# Quantification of left ventricular mass in multiple views of echocardiograms using model-agnostic meta learning in a few-shot setting

Yeong Hyeon Kim[1,*], Donghoon Kim[2,*], Jin Young Youm[3], Jiyoon Won[4], Seola Kim[5], Woohyun Park[6], Yisak Kim[1] and Dongheon Lee[1,7,8]

[1] Interdisciplinary Program in Bioengineering, Graduate School, Seoul National University, Seoul, Republic of Korea
[2] Division of Pulmonary and Allergy, Department of Internal Medicine, Chung-Ang University Hospital, Seoul, Republic of South Korea
[3] Department of Radiological Science, Nambu University, Gwangju, Republic of South Korea
[4] Department of Meridian & Acupoint, College of Korean Medicine, Dong-Eui University, Busan, Republic of South Korea
[5] Medical AI Division, Ziovision, Seoul, Republic of South Korea
[6] Department of Electrical and Computer Engineering, Sungkyunkwan University, Suwon, Republic of South Korea
[7] Department of Biomedical Engineering, Chungnam National University, Daejeon, Republic of South Korea
[8] Department of Radiology, Seoul National University College of Medicine, Seoul National University Hospital, Seoul, Republic of South Korea
* These authors contributed equally to this work.



Corresponding author
Dongheon Lee,
dhlee.jubilee@gmail.com

## ABSTRACT

**Background:** Reliable measurement of left ventricular mass (LVM) in echocardiography is essential for early detection of left ventricular dysfunction, coronary artery disease, and arrhythmia risk, yet growing patient volumes have created critical shortage of experts in echocardiography. Recent deep learning approaches reduce inter-operator variability but require large, fully labeled datasets for each standard view—an impractical demand in many clinical settings.

**Methods:** To overcome these limitations, we propose a heatmap-based point-estimation segmentation model trained *via* model-agnostic meta-learning (MAML) for few-shot LVM quantification across multiple echocardiographic views. Our framework adapts rapidly to new views by learning a shared representation and view-specific head performing K inner-loop updates, and then meta-updating in the outer loop. We used the EchoNet-LVH dataset for the PLAX view, the TMED-2 dataset for the PSAX view and the CAMUS dataset for both the apical 2-chamber and apical 4-chamber views under 1-, 5-, and 10-shot scenarios.

**Results:** As a result, the proposed MAML methods demonstrated comparable performance using mean distance error, mean angle error, successful distance error and spatial angular similarity in a few-shot setting compared to models trained with larger labeled datasets for each view of the echocardiogram.

## INTRODUCTION

The increasing number of patients with cardiovascular disease and the growing demand for echocardiography studies have led an imbalance between the availability and need for echocardiography experts (*Narang et al., 2016*). Left ventricular mass (LVM) is a strong independent predictor of adverse cardiovascular events, and an increase in LVM is associated with deterioration in left ventricular function, coronary artery disease, and an increased incidence of arrhythmias (*Bacharova et al., 2023*; *Lu et al., 2018*). Therapies aimed at reducing LVM can decrease associated risks, making reliable quantification of LVM in echocardiography essential for initial measurement, monitoring clinical response to treatments, and predicting outcomes (*Armstrong et al., 2012*; *Devereux et al., 2004*). However, compared to radiation-based medical image diagnosis, diagnosis using echocardiography is much more dependent on the operator's experience and skill (*Alsharqi et al., 2018*). Therefore, there have been reports aiming to minimize inter-observer and intra-observer variations and to assist in acquiring more consistent echocardiogram images using deep learning (*Alsharqi et al., 2018*).

Existing deep-learning approaches have demonstrated impressive accuracy and reproducibility in automating echocardiographic measurements—EchoNet-Dynamic (*Ouyang et al., 2019*), segments the left ventricle to compute ejection fraction and assess cardiomyopathy; EchoNet-LVH (*Duffy et al., 2022*), quantifies ventricular hypertrophy and tracks changes in wall thickness; and recent video-based models (*Kim et al., 2023*) classify diverse cardiac conditions from apical 4-chamber (A4C) cine loops. These fully automated methods eliminate much of the inter-observer variability inherent to manual tracing (*Duffy et al., 2022*; *Haq, Haq & Xu, 2021*; *Lang et al., 2021*), yet each evaluates only a single standard view (*e.g.*, PLAX or A4C) (*Duffy et al., 2022*; *Ouyang et al., 2019*; *Ouyang et al., 2020*).

Accurate left ventricular mass (LVM) quantification, however, requires combining linear and volumetric measurements from multiple orthogonal views—PLAX, PSAX, A2C, A4C—to capture chamber geometry fully (*Kristensen et al., 2022*; *Lang et al., 2021*; *Lu et al., 2018*; *Marwick et al., 2015*; *Mizukoshi et al., 2016*; *Takeuchi et al., 2008*). Meeting this requirement conventionally demands large, view-specific labeled datasets, placing a heavy annotation burden on clinical collaborators and limiting generalizability across institutions.

To overcome these challenges, we propose a novel, multi-view segmentation framework that leverages model-agnostic meta-learning (MAML) to learn reliable cardiac contours in a few-shot setting. By treating each echocardiographic view as a "task" within the MAML paradigm, our method dramatically reduces the need for extensive per-view annotations while preserving the measurement precision necessary for robust LVM estimation across all standard views of the echocardiography.

## MATERIALS AND METHODS

The objective of the proposed method is to locate four points in multiple views of echocardiogram (A2C, apical 2-chamber; A4C, apical 4-chamber; PLAX, parasternal long axes; and PSAX, parasternal short axes) in a few-shot environment for the quantification of

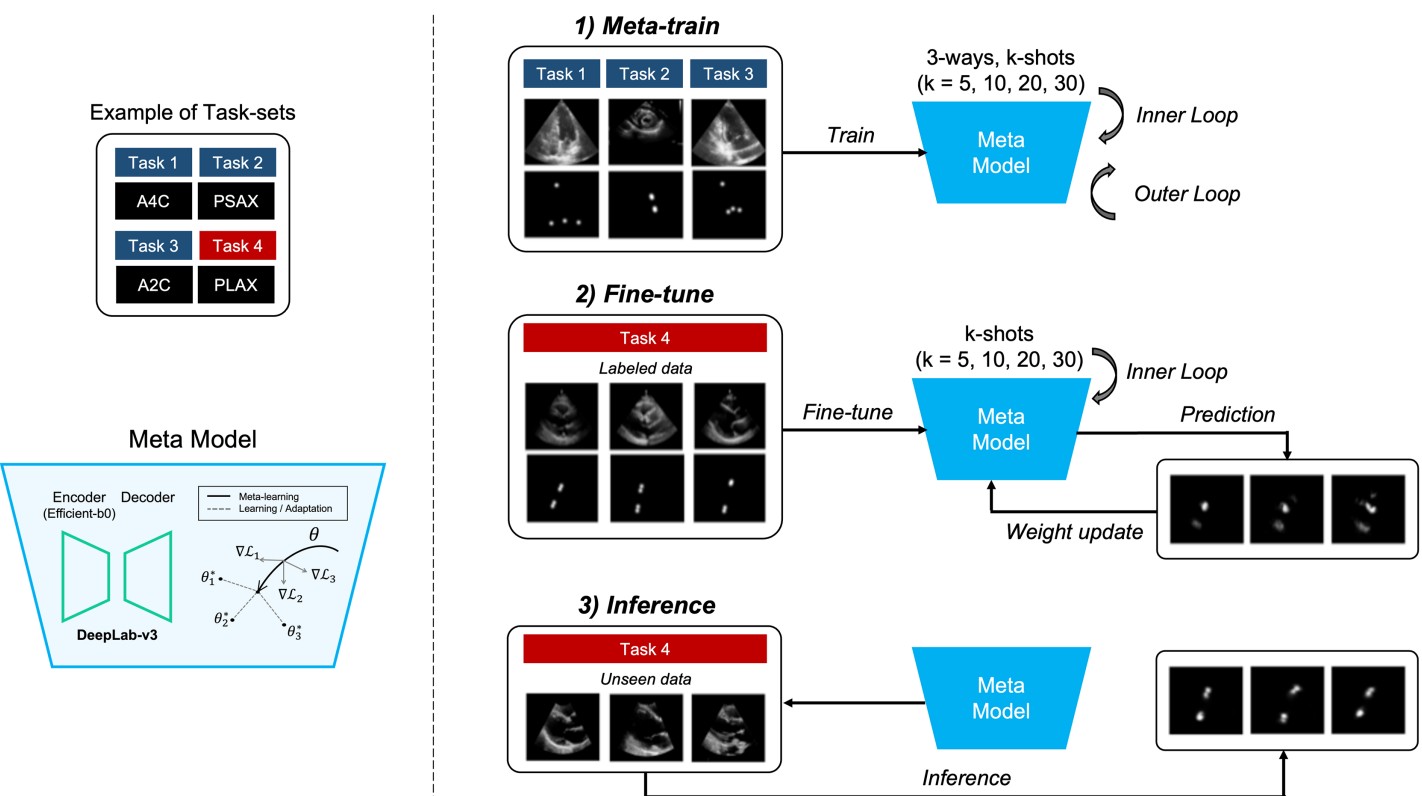

**Figure 1  Overview of our proposed method.** It represents the process of quantifying the left ventricular mass (LVM) from multiple views during echocardiography. The segmentation model with heatmap-based point estimation is utilized, and model-agnostic meta learning (MAML) methods are applied to predict the four points for LVM estimation.      

LVM (*Marwick et al., 2015*). The aim of this approach is to achieve comparable or superior performance compared to models trained on large datasets, as illustrated in detail in Fig. 1.

In this section, we will describe the segmentation method, which predicts the position of four points within each of the four views (A2C, A4C, PLAX, and PSAX) in echocardiograms. Also, training the segmentation model using several MAML methods, which consist of two stages of training, is performed. In the meta-train stage, the model is trained with data $\mathcal{D}_{tr_1}$, $\mathcal{D}_{tr_2}$ and $\mathcal{D}_{tr_3}$ by selecting from three out of four views (3-way) in each training iteration. Next, in the fine-tune stage, the other view data $\mathcal{D}_{tr_4}$ is fine-tuned with a small amount of labeled data, and the weight of the meta-model are updated. During the meta-train stage, both inner and outer iterations are performed, whereas during the fine-tune stage, only inner iterations are performed. Once the training and tuning are completed, the MAML model performs inference on unseen data $\mathcal{D}_{te_4}$ to predict points corresponding to anatomical structures in echocardiograms.

## Data preprocessing

The input images were resized to 256 × 256 pixels, and augmentation was applied with rotations ranging from 0 to 30 degrees. Additionally, the images were normalized to a range of 0 to 255 pixels, and the batch size was set to k (5, 10, 20, and 30 shots).

Annotations were conducted by both a cardiologist and an echocardiographer using the LabelMe program (*Torralba, Russell & Yuen, 2010*). All labeled data were double-checked by inter-observers, and in cases of disagreement during the validation process, annotations were revised through discussion.

## Dataset preparation

In this study, we conducted experiments on four different views of echocardiography: A2C, A4C, PLAX, and PSAX, using several open echocardiography datasets. Specifically, we utilized the publicly available EchoNet-LVH dataset (*Ouyang et al., 2019*) for the PLAX view and the TMED-2 dataset (*Huang et al., 2022*) for the PSAX view. Additionally, we utilized the publicly available CAMUS dataset (*Leclerc et al., 2019*) for both the A2C and A4C views.

The training set consisted of 100 baseline images, while the MAML method was composed of k-shot subsets with 5, 10, 20, and 30 images (Fig. A2). The validation set comprised 34 images for the A2C view, 30 images for the A4C view, 100 images for the PLAX view, and 30 images for the PSAX view. The datasets used were randomly sampled from each view's public dataset, ensuring no overlap between patients, and the test set comprised 100 images for each view.

## Semantic segmentation models

The experimented segmentation models were U-Net (*Ronneberger, Fischer & Brox, 2015*), and DeepLab-v3+ (*Chen et al., 2018*). Firstly, U-Net consists of a contracting path (encoder) to capture context and a symmetric expanding path (decoder) to enable precise localization. The architecture uses skip connections between corresponding layers in the encoder and decoder. These connections help the network retain spatial information lost during down-sampling, improving segmentation accuracy, especially for tasks that require detailed boundary predictions. In this study, the Efficientnet-B0 (*Tan & Le, 2019*) was used as the encoder model.

Additionally, DeepLab-v3 builds upon earlier versions by incorporating atrous (dilated) convolutions, which help capture multi-scale contextual information without increasing the computational cost. Atrous convolutions allow the network to enlarge the receptive field without losing resolution. DeepLab-v3 also uses a technique called the Atrous Spatial Pyramid Pooling (ASPP) module, which applies different rates of dilation to capture information at multiple scales. This architecture is particularly effective at segmenting objects in images with complex backgrounds.

## Heatmap-based point estimation in echocardiography

The segmentation models utilized the heatmap-based point estimation method (*Bulat & Tzimiropoulos, 2016*; *Zha et al., 2023*) to quantify LVM for predicting specific four points within echocardiograms. Echocardiographic images suffer from low contrast and poorly defined chamber borders, making precise landmark regression unreliable (*Dudnikov, Quinton & Alphonse, 2021*; *Mogra, 2013*). To overcome this, we convert each target point into a small Gaussian-shaped region and treat detection as a segmentation task rather than

direct coordinate regression. Concretely, for each ground-truth landmark we generate a heatmap by placing a 2D Gaussian kernel—centered at the true point and with standard deviation σ—over the surrounding pixels.

The cardiologist and the echocardiographer initially annotated the desired locations as point coordinates. From these points, a two-dimensional gaussian distribution (mask or contour) with a deviation of 7 was automatically generated and used as labels (Fig. A3). As a result, the segmentation model trained on this dataset generates a heatmap resembling a gaussian distribution. The model calculates the center points of the identified regions, ultimately determining specific point coordinates within the echocardiogram.

## Model-agnostic meta learning

MAML (*Finn, Abbeel & Levine, 2017*) is a method of meta-learning designed to learn a good set of initial parameters through various processes, enabling rapid adaptation to new tasks. MAML is not constrained by the model structure and can generally be applied to a variety of learning problems, thus delivering strong performance in tasks that require rapid adaptation with little data and only a few examples.

MAML involves a two-stage optimization process (*Finn, Abbeel & Levine, 2017*). The inner optimization starts with initial model parameters and adjusts these parameters through a few learning steps for rapid adaptation to individual tasks. The outer optimization integrates learning outcomes from various tasks to update the initial model parameters. This study applied three prominent MAML methods—first-order model-agnostic meta learning (FOMAML) (*Finn, Abbeel & Levine, 2017*), Meta-SGD (*Li et al., 2017*), and meta-curvature (*Park & Oliva, 2019*)—to multiple views of echocardiograms.

The reason for comparing these segmentation models using MAML-based few-shot learning methods is to evaluate their adaptability and performance in scenarios where data is scarce. Few-shot learning is particularly valuable in real-world applications where gathering large annotated datasets for segmentation tasks can be difficult. By using these MAML methods, the goal is to identify which variant can achieve better model generalization with fewer samples. Each MAML variant has unique properties regarding computational efficiency and adaptability, and this comparison helps pinpoint the optimal trade-offs between performance and resource usage when combined with the segmentation models.

## First-order model-agnostic meta learning

First-order model-agnostic meta learning (FOMAML) (*Finn, Abbeel & Levine, 2017*) simplifies the computational complexity in MAML by omitting the second derivative calculations that assess the impact of parameter updates on the overall model parameters after each task. This omission significantly reduces computational demands while maintaining similar performance. FOMAML approximates second-order derivatives with first-order derivatives and updates the meta-parameters with a few gradient descent steps. This makes meta-learning more practical and scalable for few-shot learning and related

tasks. At the k-th inner gradient iteration, the model parameter is $\theta_k$, the learning rate $\alpha$, and the loss $\mathcal{L}$, as shown in Eq. (1) (*Finn, Abbeel & Levine, 2017*).

$$\theta_k = \theta_{k-1} - \alpha \nabla_\theta \mathcal{L}^{(0)}(\theta_{k-1}). \tag{1}$$

During the outer iteration, the meta-objective was updated for each batch. FOMAML (*Finn, Abbeel & Levine, 2017*) simplifies the equation by omitting the second derivative term of MAML, and the initial model parameter $\theta_0$ is $\theta_{\text{meta}}$, defined as shown in Eqs. (2), (3) (*Finn, Abbeel & Levine, 2017*).

$$\theta_{meta} = \theta_{meta} - \beta g_{FOMAML} \tag{2}$$

$$g_{FOMAML} = \nabla_{\theta_k} \mathcal{L}^{(1)}(\theta_k). \tag{3}$$

## Meta-SGD

Meta-SGD (*Li et al., 2017*) enhances the flexibility and efficiency of meta-learning by adding the capability to automatically adjust learning rates through the meta-learning process. While traditional MAML methods apply the same learning rate to all parameters, Meta-SGD (*Li et al., 2017*) learns individual learning rates for each model parameter. This allows for the adjustment of learning speeds according to the importance of each parameter during the meta-learning process. Meta-SGD (*Li et al., 2017*) involves initializing and adapting the learner $f_{\theta_i}$ in the meta-space $(\theta, \alpha)$, where the learning rate $\alpha$ is typically set manually and the learner is updated through iterative gradient descent starting from a random initialization as shown in Eq. (4) (*Li et al., 2017*).

$$\theta_i = \theta_{i-1} - \alpha_{i-1} \cdot \nabla_\theta \Sigma_{\mathcal{T}_i} \mathcal{L}_{\mathcal{T}_i}(f_{\theta_{i-1}}) \tag{4}$$

In the inner iteration, weights are updated while in the outer iteration, both the meta-parameters $\theta_k$ and $\alpha_k$ are updated. The learning process is conducted with a learning rate of $\beta$ for the task distribution $p(\mathcal{T}_i)$, which defined as shown in Eqs. (5), (6) (*Li et al., 2017*).

$$\theta_k = \theta_{k-1} - \beta \nabla_\theta \Sigma_{\mathcal{T}_i \sim p(\mathcal{T}_i)} \mathcal{L}_{\mathcal{T}_i}(f_{\theta_i}) \tag{5}$$

$$\alpha_k = \alpha_{k-1} - \beta \nabla_\alpha \Sigma_{\mathcal{T}_i \sim p(\mathcal{T})} \mathcal{L}_{\mathcal{T}_i}(f_{\theta_i}) \tag{6}$$

## Meta-curvature

Meta-curvature (*Park & Oliva, 2019*) retains the basic meta-learning structure used in MAML but adds curvature information to each parameter's update, allowing for more appropriate updates for each task-specific parameter. This curvature information reflects how parameter updates are influenced by the shape of the task's loss function, enabling more sophisticated adjustments than the typical learning rates or parameter update methods. The key concept of meta-curvature (*Park & Oliva, 2019*) is to adjust the optimization path to suit the characteristics of each task. It achieves this by learning a meta-trained matrix used in parameter updates, which reflects the unique characteristics of each task's loss surface. The meta-trained matrix $M_{mc}$ can be used to expand all the same-sized meta-curvature matrices $\widehat{M}_o$, $\widehat{M}_i$ and $\widehat{M}_f$ with the Kronecker product $\otimes$,

and a $k$-dimensional identity matrix $I_k$, which defined as shown in Eqs. (7)–(9) (*Park & Oliva, 2019*).

$$\theta_i = \theta_{i-1} - M_{mc} \cdot \nabla_\theta \Sigma_{T_i} \mathcal{L}_{T_i}(f_{\theta_{i-1}}) \tag{7}$$

$$M_{mc} = \widehat{M}_o \, \widehat{M}_i \, \widehat{M}_f \tag{8}$$

$$\widehat{M}_o = M_o \otimes I_{C_{in}} \otimes I_d,$$
$$\widehat{M}_i = I_{C_{out}} \otimes M_i \otimes I_d, \tag{9}$$
$$\widehat{M}_f = I_{C_{out}} \otimes I_{C_{in}} \otimes M_f.$$

This matrix replaces the traditional scalar learning rate and allows for finer control over the direction and magnitude of parameter updates. This method enhances the adaptability and effectiveness of the learning process by aligning updates more closely with the specific needs of each task (*Park & Oliva, 2019*).

## Almost-no-inner-loop

Almost-no-inner-loop (ANIL) (*Raghu et al., 2019*) simplifies the standard MAML framework by restricting inner-loop adaptation to only the task-specific "head" parameters, while leaving the shared feature extractor fixed during task-level updates. Concretely, we partition the model parameters into a shared representation $\theta$ and a per-task head $\phi$, and perform K inner-gradient steps only on $\phi$:

$$\phi_i = \phi_{i-1} - \alpha \nabla_\phi \mathcal{L}_{T_i}(f_{\theta,\phi_{i-1}}) \tag{10}$$

with $\theta$ held constant. After K steps, the outer-loop meta-update then adjusts both $\theta$ and $\phi$ by aggregating task losses evaluated at the adapted head $\phi_k$:

$$\theta \leftarrow \theta - \beta \, \Sigma_{T_I} \nabla_\theta L_{T_I}(f_{\theta,\phi_k}),$$
$$\phi \leftarrow \phi - \beta \, \Sigma_{T_I} \nabla_\phi L_{T_I}(f_{\theta,\phi_k}). \tag{11}$$

By eliminating inner-loop updates on $\theta$, ANIL achieves a dramatic reduction in per-task computation and highlights that rapid adaptation primarily occurs in the final layer, while the shared representation learned $\theta$ generalizes across tasks. An illustration of the differences between the MAML and ANIL methods is shown in Supplemental Fig. A4.

## Implementation details

The experimented segmentation models were trained using the mean square error (MSE) loss function with the Adam optimizer, employing a learning rate of 5e−3. During the meta-training process, 50 epochs were applied, and the learning rate between adaptations in meta-training was set to 0.03. Adaptation was performed 10 times for all shots, and the learning rate between meta-tests was set to 0.05, with 100 adaptation steps. Additionally, the final model was selected based on the highest mean pixel accuracy observed in the validation set. In the inference process, the highest probability value in the region was extracted as a point and used as the final predicted value. To compare the performance, the baseline was set to 100-shot, and the MAML methods were varied by 5, 10, 20, and 30 shots.

The development was implemented using PyTorch (ver. 1.12.1) and learn2learn (ver. 0.1.7), and the model was trained using NVIDIA RTX A6000 GPU on Linux (Ubuntu 18.04). Our implementation is available here: https://github.com/KimYeongHyeon/Metalearning-echocardiogram. Here, the pre-trained model weights are also made available, along with dataset selected and labeled from the raw public data. Specifically, EchoNet-LVH dataset (https://echonet.github.io/lvh/index.html#dataset) (*Ouyang et al., 2019*) is used for the PLAX view, the TMED-2 dataset (https://tmed.cs.tufts.edu/tmed_v2.html) (*Huang et al., 2022*) for the PSAX view, and the CAMUS dataset (https://www.creatis.insa-lyon.fr/Challenge/camus/databases.html) (*Leclerc et al., 2019*) for both the A2C and A4C views. These resources are provided for the MAML experiments in this study.

## Evaluation metrics

The proposed method was quantitatively evaluated using the mean distance error (MDE), which calculates the difference between the reference points and the coordinates predicted by the model. Additionally, we introduced a new evaluation metric, successful detection rate (SDR), which evaluates whether predicted points fall within a certain threshold distance. Furthermore, the model predicted line segments formed by the points such as the intraventricular septum (IVS), left ventricular internal dimension (LVID), and left ventricular posterior wall (LVPW) in the PLAX view (*Kristensen et al., 2022*). The mean angle error (MAE), which measures the angle difference between the predicted line and the reference line, was also evaluated.

Furthermore, we introduce a novel evaluation metric, spatial angular similarity (SAS), which jointly captures both the degree of parallelism (angular similarity) and the distance difference between two lines by integrating elements of SDR and MAE. The metric is defined as:

$$SAS = 100 \times (\alpha \cdot |cos\theta| + (1 - \alpha) \cdot e^{-\beta d}). \tag{12}$$

Here, $\alpha$ (ranging from 0 to 1) weights the orientation component, d denotes the distance between the two lines, $\beta$ determines the rate at which the distance-based score decays, and perfectly parallel lines are guaranteed a minimum score of $\alpha \times 100$ regardless of their separation.

In clinical practice, small errors in landmark localization can lead to substantial inaccuracies in left ventricular mass, since LVM formulas combine linear measurements (*e.g.*, septal thickness, cavity diameter) in three-dimensional measures. Our suite of metrics therefore provides a multi-faceted assessment of how well the model meets the precision demands of LVM quantification.

## RESULTS

### Performance comparison of segmentation models

In this study, we utilized the U-Net (*Ronneberger, Fischer & Brox, 2015*), DeepLab-v3+ (*Chen et al., 2018*) and SegFormer (*Xie et al., 2021*) models along with the heatmap-based point estimation method (*Bulat & Tzimiropoulos, 2016*; *Zha et al., 2023*) to quantify LVM

**Table 1 Overview of our proposed method: It represents the process of quantifying the left ventricular mass (LVM) from multiple views during echocardiography.** The segmentation model with heatmap-based point estimation is utilized, and model-agnostic meta learning (MAML) methods are applied to predict the four points for LVM estimation.

| Segmentation model | Dice coefficient ↑ | Mean pixel accuracy ↑ | Mean distance error ↓ | Spatial angular similarity ↑ |
|---|---|---|---|---|
| U-Net (*Ronneberger, Fischer & Brox, 2015*) | 0.597 | 0.98 | 5.882 | 75.689 |
| DeepLab-v3+ (*Chen et al., 2018*) | 0.548 | 0.989 | 5.624 | 74.189 |
| SegFormer (*Xie et al., 2021*) | 0.196 | 0.98 | 5.514 | 75.331 |

by predicting specific four points within echocardiograms. We compared the performance of these two models under the baseline setting (100-shot) and used the dice coefficient and mean pixel accuracy as evaluation metrics.

Table 1 presents the results of evaluating four models using the Dice coefficient, mean pixel accuracy, mean distance error and spatial angular similarity metrics. The similarity in performance suggests that the model type does not significantly impact the outcomes, and the lower results from the dice coefficient metric—especially for SegFormer—can be attributed to training with a mean square error loss function on gaussian distribution labels in heatmap form rather than binary masks. Based on the balance of accuracy and localization, DeepLab-v3+ (*Chen et al., 2018*), which represents superior performance, was selected as the baseline model for subsequence metal-learning experiments.

## Performance comparison of MAML methods

We implemented a segmentation model, DeepLab-v3+ (*Chen et al., 2018*) which demonstrated the best performance of mean pixel accuracy, to locate specific four points within an echocardiogram using several MAML methods, including FOMAML (*Finn, Abbeel & Levine, 2017*), Meta-SGD (*Li et al., 2017*), Meta-Curvature (*Park & Oliva, 2019*) and Almost-No-Inner-Loop (ANIL) (*Raghu et al., 2019*). The quantitative evaluation of the MAML methods is presented in Table 2 and Supplemental Tables A1–A6, showing the results of mean distance error (MDE), mean angle error (MAE) and SAS in a few-shot environment from multiple views of echocardiogram. It compares the performance of MAML methods to that of a baseline (100-shot) trained with more labels. As the k-shot increased, the errors of the MAML methods decreased, and when the k-shot reached 30, they exhibited similar or better performance compared to the baseline ($p > 0.05$). Specifically, in the A2C, A4C, and PLAX views, FOMAML (*Finn, Abbeel & Levine, 2017*) demonstrated the highest performance; however, there is no statistical difference in performance ($p > 0.05$). For the PSAX view, the proposed FOMAML (*Finn, Abbeel & Levine, 2017*) demonstrated statistically significantly higher performance compared to the baseline in terms of the MAE metric ($p = 0.0493$). The k-shot was low, the performance differences among the MAML methods, but as it increased, the performance saturated.

Figure 2 illustrates the results of successful detection rate (SDR) as an extended evaluation of MDE. It indicates the prediction rate of points within the echocardiogram with the pixel-level threshold varied. The k-shot was set to 30, as the threshold increased, the performance of the MAML methods showed similar or better performance compared to the baseline.

**Table 2 The quantitative results for each view of echocardiogram using several model-agnostic meta learning (MAML) methods were compared.** The performance of the baseline (100-shot) and MAML methods was evaluated based on mean distance error (MDE) and mean angle error (MAE) and spatial angular similarity (SAS).

| k-shot | Model | Metric | A2C (CAMUS *Leclerc et al., 2019*) | A4C (CAMUS *Leclerc et al., 2019*) | PLAX (EchoNet-LVH *Ouyang et al., 2019*) | PSAX (TMED-2 *Huang et al., 2022*) |
|---|---|---|---|---|---|---|
| 100 | Baseline | MDE | 4.56 ± 3.84 | 5.17 ± 5.23 | 5.11 ± 3.96 | 7.38 ± 5.67 |
| | | MAE | – | – | 10.25 ± 10.70 | 24.65 ± 26.03 |
| | | SAS | 78.14 ± 6.67 | 77.52 ± 8.80 | 77.40 ± 8.09 | 64.10 ± 11.47 |
| 5 | FOMAML (*Finn, Abbeel & Levine, 2017*) | MDE | 13.62 ± 14.36 | 12.14 ± 11.22 | 7.39 ± 8.13 | 10.87 ± 7.67 |
| | | MAE | – | – | 16.52 ± 17.42 | 23.98 ± 22.14 |
| | | SAS | 64.03 ± 14.00 | 66.10 ± 11.35 | 70.04 ± 9.42 | 58.98 ± 10.58 |
| | Meta-SGD (*Li et al., 2017*) | MDE | 12.09 ± 14.82 | 12.55 ± 13.64 | 6.86 ± 4.68 | 12.72 ± 11.47 |
| | | MAE | – | – | 19.74 ± 26.87 | 30.14 ± 32.03 |
| | | SAS | 66.39 ± 14.33 | 65.01 ± 9.87 | 68.21 ± 9.79 | 56.24 ± 12.71 |
| | Meta-Curvature (*Park & Oliva, 2019*) | MDE | 10.62 ± 15.08 | 10.85 ± 10.73 | 6.99 ± 5.40 | 10.24 ± 7.16 |
| | | MAE | – | – | 17.39 ± 17.96 | 23.72 ± 21.16 |
| | | SAS | 69.73 ± 11.62 | 67.47 ± 9.13 | 69.21 ± 8.77 | 60.43 ± 10.30 |
| | ANIL (*Raghu et al., 2019*) | MDE | 39.33 ± 39.01 | 49.33 ± 45.50 | 15.60 ± 13.92 | 13.21 ± 11.64 |
| | | MAE | 63.88 ± 53.97 | 33.60 ± 38.49 | 32.89 ± 37.41 | 24.58 ± 35.07 |
| | | SAS | 45.02 ± 11.55 | 50.38 ± 12.46 | 54.24 ± 12.35 | 59.15 ± 12.86 |
| 10 | FOMAML (*Finn, Abbeel & Levine, 2017*) | MDE | 8.66 ± 8.39 | 8.39 ± 7.68 | 7.62 ± 7.58 | 11.06 ± 11.21 |
| | | MAE | – | – | 18.57 ± 21.88 | 28.50 ± 32.42 |
| | | SAS | 69.80 ± 7.40 | 70.80 ± 7.35 | 70.39 ± 8.96 | 59.45 ± 9.62 |
| | Meta-SGD (*Li et al., 2017*) | MDE | 10.35 ± 12.05 | 7.81 ± 7.53 | 6.76 ± 5.18 | 9.15 ± 7.08 |
| | | MAE | – | – | 12.59 ± 14.65 | 21.85 ± 20.53 |
| | | SAS | 68.56 ± 8.91 | 72.53 ± 7.17 | 71.93 ± 8.64 | 64.03 ± 10.10 |
| | Meta-Curvature (*Park & Oliva, 2019*) | MDE | 11.37 ± 15.49 | 9.29 ± 8.61 | 6.04 ± 4.67 | 9.87 ± 7.96 |
| | | MAE | – | – | 10.48 ± 13.89 | 18.34 ± 16.48 |
| | | SAS | 66.97 ± 12.15 | 68.58 ± 9.44 | 74.22 ± 7.62 | 63.89 ± 10.20 |
| | ANIL (*Raghu et al., 2019*) | MDE | 37.19 ± 38.56 | 37.26 ± 35.47 | 13.97 ± 13.03 | 16.70 ± 13.10 |
| | | MAE | 57.12 ± 56.28 | 51.91 ± 49.80 | 37.20 ± 41.25 | 29.53 ± 33.23 |
| | | SAS | 47.54 ± 11.82 | 44.91 ± 14.63 | 55.98 ± 9.90 | 54.46 ± 10.64 |
| 20 | FOMAML (*Finn, Abbeel & Levine, 2017*) | MDE | 6.84 ± 4.38 | 6.24 ± 4.18 | 5.44 ± 3.67 | 9.01 ± 7.44 |
| | | MAE | – | – | 10.71 ± 9.32 | 18.66 ± 19.10 |
| | | SAS | 71.85 ± 6.84 | 73.46 ± 6.70 | 75.61 ± 6.54 | 64.83 ± 9.75 |
| | Meta-SGD (*Li et al., 2017*) | MDE | 8.49 ± 7.80 | 6.99 ± 7.23 | 5.81 ± 4.34 | 8.86 ± 6.99 |
| | | MAE | – | – | 8.45 ± 7.79 | 19.82 ± 19.85 |
| | | SAS | 69.54 ± 11.32 | 73.49 ± 7.93 | 75.27 ± 8.50 | 64.70 ± 9.81 |
| | Meta-Curvature (*Park & Oliva, 2019*) | MDE | 9.34 ± 6.87 | 7.43 ± 7.17 | 5.81 ± 5.74 | 9.16 ± 5.73 |
| | | MAE | – | – | 10.07 ± 15.20 | 18.34 ± 16.30 |
| | | SAS | 69.93 ± 7.86 | 72.24 ± 8.56 | 76.02 ± 7.65 | 63.55 ± 8.62 |
| | ANIL (*Raghu et al., 2019*) | MDE | 34.95 ± 36.93 | 20.98 ± 23.08 | 14.56 ± 14.86 | 16.37 ± 13.49 |
| | | MAE | 40.59 ± 45.17 | 22.27 ± 30.41 | 29.53 ± 38.05 | 38.45 ± 35.78 |
| | | SAS | 49.21 ± 14.33 | 60.65 ± 13.04 | 58.88 ± 11.59 | 49.46 ± 10.95 |

| k-shot | Model | Metric | A2C (CAMUS *Leclerc et al., 2019*) | A4C (CAMUS *Leclerc et al., 2019*) | PLAX (EchoNet-LVH *Ouyang et al., 2019*) | PSAX (TMED-2 *Huang et al., 2022*) |
|---|---|---|---|---|---|---|
| 30 | FOMAML (*Finn, Abbeel & Levine, 2017*) | MDE | 5.48 ± 4.28 | 5.45 ± 6.30 | 5.06 ± 3.24 | 7.88 ± 4.46 |
| | | MAE | – | – | 7.67 ± 6.64 | 15.23 ± 12.01 |
| | | SAS | 77.01 ± 5.40 | 77.70 ± 7.60 | 77.36 ± 7.37 | 67.23 ± 8.32 |
| | Meta-SGD (*Li et al., 2017*) | MDE | 5.87 ± 4.75 | 5.69 ± 5.84 | 5.45 ± 3.68 | 7.11 ± 3.80 |
| | | MAE | – | – | 8.57 ± 11.96 | 13.53 ± 10.13 |
| | | SAS | 76.39 ± 4.98 | 77.23 ± 8.57 | 75.84 ± 7.22 | 69.19 ± 7.96 |
| | Meta-Curvature (*Park & Oliva, 2019*) | MDE | 6.42 ± 3.92 | 6.04 ± 7.50 | 5.56 ± 4.28 | 7.78 ± 4.57 |
| | | MAE | – | – | 8.16 ± 10.68 | 12.73 ± 10.36 |
| | | SAS | 72.83 ± 5.66 | 76.46 ± 9.67 | 75.91 ± 7.86 | 68.24 ± 8.07 |
| | ANIL (*Raghu et al., 2019*) | MDE | 29.70 ± 35.43 | 16.12 ± 21.74 | 11.13 ± 10.99 | 13.51 ± 13.00 |
| | | MAE | 40.95 ± 47.34 | 17.70 ± 28.13 | 27.47 ± 27.14 | 33.38 ± 38.27 |
| | | SAS | 53.41 ± 14.03 | 64.69 ± 14.12 | 59.80 ± 10.66 | 56.41 ± 14.63 |

**Note:**
A2C, apical 2-chamber; A4C, apical 4-chamber; PLAX, parasternal long axes; PSAX, parasternal short axes; MDE: mean distance error; MAE: mean angle error angle error (MAE), SAS: spatial angular similarity.

Figure 3 presents qualitative results comparing the ground truth with the prediction results in the baseline (100-shot) and the FOMAML (*Finn, Abbeel & Levine, 2017*) method with a k-shot of 30. In the A2C and A4C views, the reference points are represented by red points, while the predicted results are depicted as green square points (Figs. 3A, 3B). Additionally, in the PLAX and PSAX views, the reference lines are represented by red points and lines (IVS, LVID, and LVPW), while the predicted results are shown as green square points and lines.

## DISCUSSION

The demand for echocardiography experts is increasing due to the rising number of cardiovascular disease patients and growing prevalence of echocardiography studies (*Narang et al., 2016*). Quantitatively measuring left ventricular mass (LVM) in echocardiograms is clinically crucial (*Armstrong et al., 2012*; *Bacharova et al., 2023*; *Devereux et al., 2004*; *Kim et al., 2022*; *Lu et al., 2018*); however, echocardiography diagnosis heavily depends on individual experience, resulting in significant subjectivity in image interpretation (*Alsharqi et al., 2018*).

In clinical practice, left ventricular hypertrophy (LVH) is diagnosed when LVM indexed to body surface area exceeds guideline thresholds (*e.g.*, >115 g/m$^2$ in men or >95 g/m$^2$ in women), making precise LVM measurement indispensable for LVH detection and grading (*Lang et al., 2021*). Large prospective studies have shown that every 50 g increase in LVM confers roughly a 20% higher risk of heart failure, stroke, atrial fibrillation and all-cause mortality, while regression of LVM through antihypertensive or other targeted therapies is directly linked to improved survival and fewer cardiovascular events (*Armstrong et al., 2012*; *Devereux et al., 2004*).

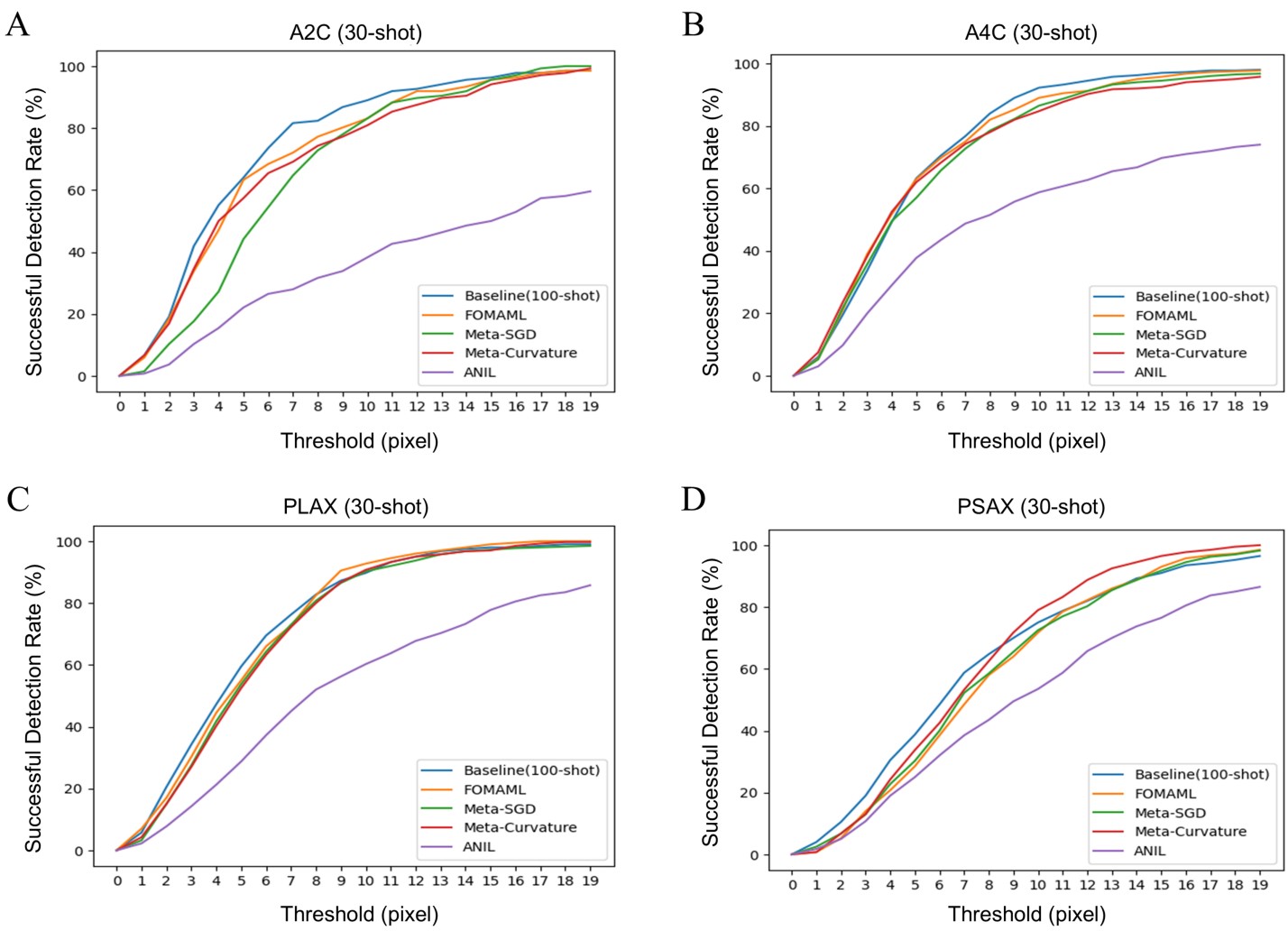

**Figure 2** The results of successful detection rate (SDR) for each view of echocardiogram using several model-agnostic meta learning (MAML) methods were analyzed. The k-shot was set to 30, demonstrating the performance variation of the model using different threshold. (A) A2C view (B) A4C view (C) PLAX view (D) PSAX view.

Methods utilizing deep learning on echocardiograms to calculate clinically important indicators in the left ventricular chamber (*Duffy et al., 2022*; *Ouyang et al., 2019*; *Ouyang et al., 2020*) have addressed these variations and achieved more standardized image acquisition and interpretation (*Alsharqi et al., 2018*). However, measurements from multiple views are essential for accurate quantification of LVM in echocardiograms, thus requiring labeled data for each view (*Duffy et al., 2022*; *Ouyang et al., 2019*; *Ouyang et al., 2020*). The model-agnostic meta-learning (MAML) method, utilizing limited labeled data for training, has been applied across various medical domains (*Godau & Maier-Hein, 2021*), including chest X-ray images (*Naren, Zhu & Wang, 2021*), 3D CT data (*Lachinov, Getmanskaya & Turlapov, 2020*), and skin images (*Khadka et al., 2022*), but has not been employed for multiple views of echocardiograms.

In contrast to existing deep-learning pipelines that train separate, fully supervised models on large, single-view datasets, our method reframes multi-view LVM estimation as

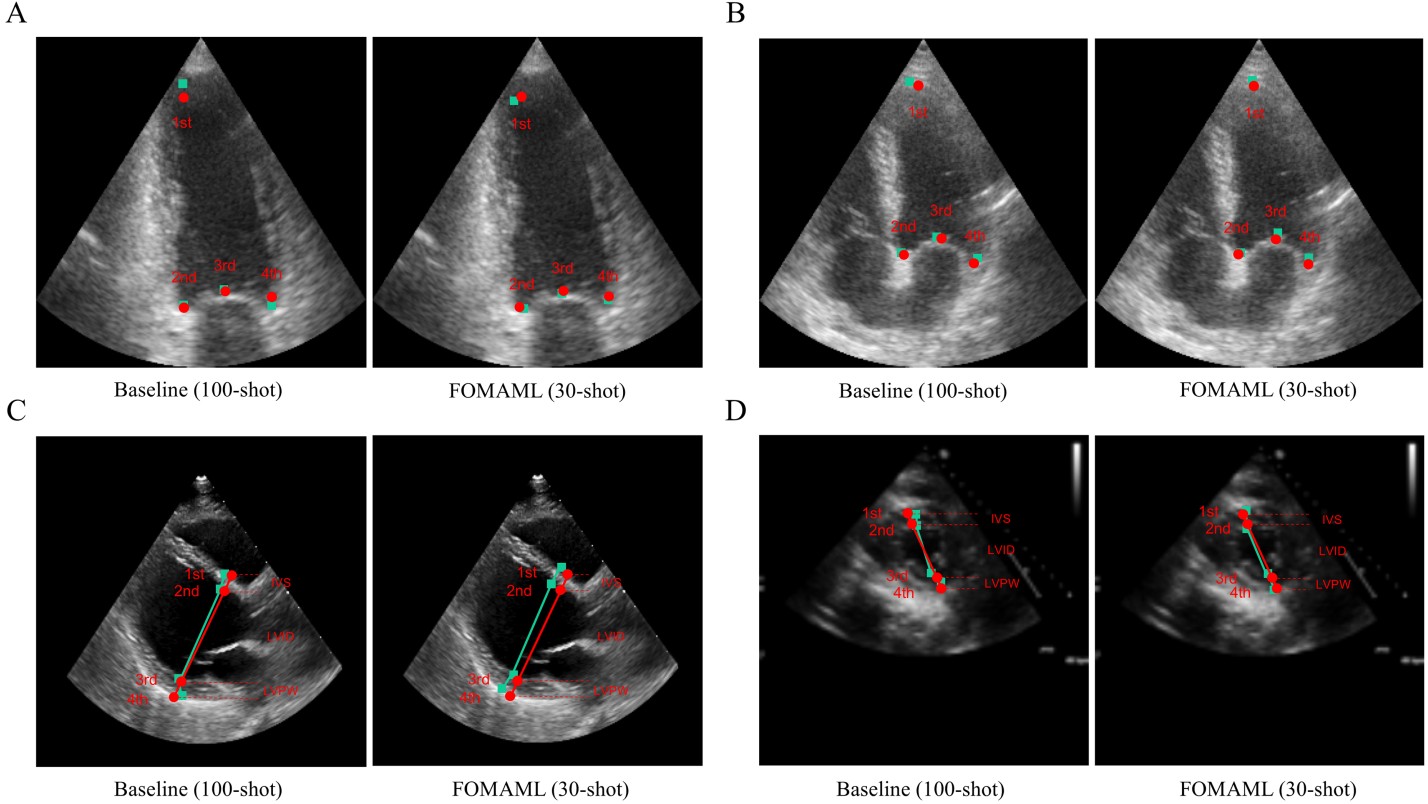

**Figure 3** The qualitative results are represented for the baseline (100-shot) and FOMAML (30-shot) in the following views. (A, B) A2C and A4C views; where the ground truth is indicated by red points, while the predicted results are indicated as green square points. (C, D) PLAX and PSAX views; where the ground truth is indicated by red points and lines (IVS, LVID, and LVPW), and the predicted results are indicated as green square points and lines.

a meta-learning problem. By treating each standard echocardiographic view (A2C, A4C, PLAX, PSAX) as an individual "task" within the MAML framework, we learn a single shared initialization that captures the common anatomical priors across views—dramatically reducing the hundreds of labels normally required per view to as few as 5–30 shots. To address the inherently ambiguous chamber boundaries in 2D echoes (*Dudnikov, Quinton & Alphonse, 2021*; *Mogra, 2013*), we predict four critical LVM landmarks *via* heatmap-based Gaussian point estimation rather than binary masks. In k-shot experiments (k = 5, 10, 20, 30), our MAML-adapted DeepLab-v3+ matches or exceeds the 100-shot baseline across distance, angular, and Dice metrics, demonstrating efficient cross-view generalization. To our knowledge, this is the first study to (1) detect the specific point landmarks required for LVM formulae and (2) validate the effectiveness of model-agnostic meta-learning for few-shot, multi-view quantification in echocardiography.

It is known that our proposed method can calculate LVM from various multiple views (Supplemental Fig. A1). LVM can be calculated from the PLAX view using the Devereux formula (Supplemental Figs. A1–A1F), and another method for LVM calculation is the area-length method (A-L) which can be applied in the A4C view to calculate a (apex to

short-axis-plane) + d (short-axis-plane to mitral-plane) (Supplemental Figs. A1–A1E). Additionally, utilizing an additional segmentation model, A1 (epicardial area), A2 (endocardial area), and t (A1–A2, wall thickness) can be obtained from the PSAX view, enabling complete LVM calculation. Moreover, to calculate LVM using the biplane model (BP) in the A4C and A2C views, it is necessary to determine multiple coordinates of the boundary plane (Supplemental Figs. A1–A1C). Through our proposed method, we predicted the coordinates of the boundary planes at the main locations, demonstrating the ability to calculate LVM using various methods.

From a theoretical standpoint, the convergence of all MAML methods toward baseline performance by k = 30 reflects that the meta-learner successfully captures a low-dimensional manifold of echocardiographic contour functions shared across views, such that only minimal fine-tuning is required for each new task. This rapid adaptation aligns with the MAML objective of finding an initialization that lies close to every task's optimum in parameter space, effectively reducing the expected adaptation error with few gradient steps. The fact that different backbones—U-Net (*Ronneberger, Fischer & Brox, 2015*), DeepLab-v3+ (*Chen et al., 2018*), SegFormer (*Xie et al., 2021*)—perform similarly under the same heatmap-based loss further suggests that the meta-learned initialization encodes anatomy-specific priors that dominate over individual architecture inductive biases Additionally, because ANIL (*Raghu et al., 2019*) uses the feature extractor learned during meta-training as a fixed component, it cannot properly extract features for novel views, leading to relatively poorer performance compared to other MAML methods.

Compared to fully supervised learning, which requires hundreds of labeled examples per view, MAML-based few-shot learning reduces the annotation burden by an order of magnitude while achieving comparable—or even superior—accuracy, cutting expert labeling costs and time in clinical practice and enabling rapid adaptation to new views or patient populations with minimal additional data. Altogether, this demonstrates that multi-view echocardiographic quantification can be framed as a shared meta-learning problem—where anatomical commonalities drive few-shot generalization—and that MAML offers a scalable, annotation-efficient path toward AI-driven cardiac imaging.

To quantitatively evaluate the results of our proposed segmentation model using MAML, we introduced a new metric called successful detection rate (SDR) in addition to mean distance error (MDE). This is because there are no absolute reference points within the echocardiogram in terms of their starting or ending points for LVM estimation. Additionally, we evaluated using the mean angle error (MAE) metric to overcome the limitations of MDE and SDR metrics, as the positions of structures such as IVS, LVID, and LVPW may vary between healthcare providers, particularly in the PLAX view. Therefore, instead of evaluating solely based on the coordinates of points, we assessed the gradient between the predicted lines (IVS, LVID, and LVPW) and the reference lines (Supplemental Tables A1–A6).

This study had several limitations. Firstly, although various types of echocardiogram open datasets were utilized, they did not provide information in millimeter units, necessitating a time-consuming standardization process and expert labeling for the segmentation task of this study. In addition, the inclusion of datasets with relatively low

resolution, such as the A2C view (*e.g.*, CAMUS *Leclerc et al., 2019*), might have contributed to a decrease in model performance. Therefore, in the future, it is necessary to enhance the quality of data by utilizing high-resolution real-world data used in clinical settings, which can be quantitatively evaluated in millimeter units. Moreover, addressing the data bias issue through multi-institutional validation will be crucial for improving model performance. Secondly, as the points used for LVM estimation lack fixed positions in the echocardiogram regarding their starting or ending points, we introduced a novel evaluation metric called the successful detection rate (SDR). This metric measures if predicted points are within a specified threshold distance, providing a solution to the challenges posed by the mean distance error (MDE) metric in echocardiograms. Finally, since linear and volumetric measurements of LVM in multiple views are calculated based on formulas, minor measurement inaccuracies can result in significant errors (*Kristensen et al., 2022*). Therefore, for practical clinical application, the performance of the segmentation model needs to be refined to ensure more precise measurements.

## CONCLUSIONS

In this work, we proposed a segmentation model to quantify LVM using MAML methods in echocardiograms from multiple views. Our method demonstrated the ability to effectively identify points crucial for LVM calculation even with limited data in multiple views. To advance toward clinical deployment, future work will focus on incorporating high-resolution, clinically calibrated cine datasets with multi-center validation, extending the framework to model spatio-temporal continuity in echocardiographic loops.

## ACKNOWLEDGEMENTS

We would like to express our sincere gratitude to the 'Korea Human Resource Development Institute for Health & Welfare (KOHI)' for providing the educational courses that were instrumental in conducting this research. I used ChatGPT o3-mini for English proofreading.

### Funding

This work was supported by Chungnam National University (2023-0540-01) and Chungnam National University Hospital (2021-2122-01). The funders had no role in study design, data collection and analysis, decision to publish, or preparation of the manuscript.

### Grant Disclosures

The following grant information was disclosed by the authors:
Chungnam National University: 2023-0540-01.
Chungnam National University Hospital: 2021-2122-01.

### Competing Interests

Seola Kim is employed by Ziovision.

## Author Contributions

- Yeong Hyeon Kim conceived and designed the experiments, performed the experiments, performed the computation work, prepared figures and/or tables, authored or reviewed drafts of the article, and approved the final draft.
- Donghoon Kim conceived and designed the experiments, performed the computation work, prepared figures and/or tables, authored or reviewed drafts of the article, and approved the final draft.
- Jin Young Youm analyzed the data, prepared figures and/or tables, and approved the final draft.
- Jiyoon Won analyzed the data, prepared figures and/or tables, and approved the final draft.
- Seola Kim performed the experiments, prepared figures and/or tables, and approved the final draft.
- Woohyun Park performed the experiments, prepared figures and/or tables, and approved the final draft.
- Yisak Kim performed the experiments, prepared figures and/or tables, and approved the final draft.
- Dongheon Lee conceived and designed the experiments, performed the computation work, prepared figures and/or tables, authored or reviewed drafts of the article, and approved the final draft.

## Data Availability

The data and code are available in the Supplemental Files, and Zenodo:

Kim, Y., Kim, D., Youm, J. Y., & Won, J. (2024). Quantification of Left Ventricular Mass in Multiple Views of Echocardiograms using Model-agnostic Meta Learning in a Few-Shot Setting. In PeerJ Computer Science. Zenodo. https://doi.org/10.5281/zenodo.16794861.

The EchoNet-LVH is available at: https://echonet.github.io/lvh/index.html#dataset.

The Tufts Medical Echocardiogram Dataset (TMED) is available at: https://tmed.cs.tufts.edu/tmed_v2.html.

The CAMUS dataset is available at: https://www.creatis.insa-lyon.fr/Challenge/camus/databases.html.

## Supplemental Information

Supplemental information for this article can be found online at http://dx.doi.org/10.7717/peerj-cs.3161#supplemental-information.

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
