# Peer review of "Quantification of left ventricular mass in multiple views of echocardiograms using model-agnostic meta learning in a few-shot setting"

_PeerJ Computer Science, doi:10.7717/peerj-cs.3161_

## Round 0.1 · original submission · Major Revisions

Given the evaluations of three expert reviewers, the manuscript deserves a major revision, as motivated below.

The manuscript addresses a relevant topic and proposes a method for segmenting echocardiographic images with a model-agnostic meta-learning approach for few-shot learning. It shows competitive results w.r.t. baseline models.

However, all reviewers are concerned about the limited discussion of the state of the art (other models should be reported) and the lack of comparison with other competitive models.

Therefore, a major review is assigned to make the authors improve their manuscript in the abovementioned critical aspects, as well as in other minor issues that the reviewers raised (see reviewers' comments below).

Reviewer 1 ·

Basic reporting

This paper presents a novel approach for quantifying left ventricular mass (LVM) in echocardiograms using model-agnostic meta-learning (MAML) in a few-shot setting. The proposed method addresses the challenge of requiring large amounts of labeled data for multiple echocardiogram views by leveraging MAML-based segmentation models. These models employ heatmap-based point estimation to predict specific anatomical points within echocardiograms, enabling LVM quantification with limited data. Experimental results demonstrate that the MAML approach achieves comparable performance to models trained with larger labeled datasets, measured through metrics like mean distance error, mean angle error, and successful distance error.

The study tackles the issue of operator dependency and subjectivity in echocardiography interpretation, offering a scalable and data-efficient solution for reliable LVM measurement across multiple views. This work could significantly contribute to improving the automation and consistency of echocardiographic analysis in clinical settings.

Experimental design

The use of mean distance error, mean angle error, and successful distance error as performance metrics provides a robust framework to evaluate the accuracy and reliability of the proposed segmentation model. Moreover, the few-shot learning environment is well-designed to test the model’s ability to generalize with limited labeled data, reflecting real-world scenarios where data for multiple echocardiogram views may be scarce. Lastly, The experiments are conducted across multiple views of echocardiograms, showcasing the generalizability of the model and its ability to handle variations in input perspectives.

Validity of the findings

The results clearly indicate that the proposed MAML-based method performs well in a few-shot setting, achieving comparable accuracy to fully supervised models. This quantitative evidence supports the effectiveness of the approach.

Additional comments

The experimental design could be further improved. For example, although the baselines include FOMAML and Meta-SGD, which represent competitive performance, the overall baseline do not include more recent methods. The authors should consider more recent baselines with better performance to demonstrate the validity of the proposed method.

·

Basic reporting

In general, the topic invested in this paper is interesting and meets the scope of the journal. Experimental results confirm the system's effectiveness . The paper is well written, and the results seem to be reasonable. The authors should revise the paper to further improve its quality before I vote for an acceptance. My comments are as follows:
1- Author must keep the abstract in the following format:
a) Introduction about the requirement of proposed work.
b) Drawbacks of existing work.
c) The proposed work.
d) Which dataset is used.
2- In introduction, author should mention specifically the one paragraph of drawbacks of existing work and how the proposed methodology resolves that problem. To be more precise author can add one tabular format of existing work.

3- The literature review needs to be improved (length, comprehension, etc.).

4- A deeper theoretical analysis is also indispensable. In the experimental part, the authors seem to simply take their results and compare them with some results from the literature. Numerically, their method has an advantage. However, the reasons for these advances need to be further explored.

5- In section conclusion, discuss the weaknesses of this model and describe your future goals and how you would extend the proposed algorithm for such problems.

Experimental design

4- A deeper theoretical analysis is also indispensable. In the experimental part, the authors seem to simply take their results and compare them with some results from the literature. Numerically, their method has an advantage. However, the reasons for these advances need to be further explored

Validity of the findings

Validated-Good

Additional comments

Please revise manuscript as per the comments

Reviewer 3 ·

Basic reporting

The authors should explain in more details the impact of why accurate LVM quantification is essential. Explain the gap in existing literature and how the authors' approach can handle it.
Simplify technical terms for a broader audience.
The authors may want to include more figures to describe the approach.

Experimental design

The approach presented by the authors is technically sound given its use of MAML for few-shot learning and segmentation models such as U-Net and DeepLab-v3+. The methods are suitable for the problem of echocardiogram segmentation in a few-shot learning environment.
However, the authors should consider to describe more on the models and techniques:
- While U-Net and DeepLab-v3+ are widely used, the authors should explain why they are optimal for the specific task of heatmap-based point estimation in echocardiograms. The authors should compare with other models would to help demonstrate why what they do is better.
- Evaluation Metrics: While metrics eg. mean distance error and distance error are mentioned, a more detailed explanation of how these metrics directly affect model performance and clinical relevance (in terms of LVM quantification) would be good. It would be helpful to include tests on other models to highlight the performance differences more clearly.
- The manuscript focuses on the MAML framework for training with limited data, but it would benefit from a discussion on how the few-shot models compare to models trained on larger datasets. A more detailed analysis of the generalizability and efficiency of MAML-based models would add a practical element to the research.

Validity of the findings

The authors used diverse datasets and talked about the SDR and MAE, which are good for the challenges in echocardiogram analysis. However the authors should compare with more approaches.

---

## Round 0.2 · accepted · Accept

You have successfully addressed the concerns raised by the reviewers, and the manuscript is now suitable for acceptance.

·

Basic reporting

no comments.

Experimental design

The paper is well-structured and focuses on a highly relevant topic. Due to its interdisciplinary nature and broad readership, it would be beneficial to introduce definitions for some technical terms . I recommend to accept.

Validity of the findings

The paper presents an innovative and integrated approaches. The conclusions are valid, Recommended for acceptance.

Additional comments

Recommended